# Association of serum 25-Hydroxy vitamin D with total and regional adiposity and cardiometabolic traits

Reka Karuppusami[1]☯, Belavendra Antonisami[1], Senthil K. Vasan[2]☯, Mahasampath Gowri[1], Hepsy Y. Selliah[1], Geethanjali Arulappan[3], Felix Jebasingh[4], Nihal Thomas[4], Thomas V. Paul☯[4] *

**1** Department of Biostatistics, Christian Medical College, Vellore, India, **2** Oxford Centre for Diabetes, Endocrinology and Metabolism, Churchill Hospital, University of Oxford, Oxford, United Kingdom, **3** Department of Clinical Biochemistry, Christian Medical College, Vellore, India, **4** Department of Endocrinology, Diabetes and Metabolism, Christian Medical College, Vellore, India

☯ These authors contributed equally to this work.
* thomasvpaul@yahoo.com

**Data Availability Statement:** All relevant data are within the paper and its Supporting Information files.

## Abstract

### Background

Lower serum 25-hydroxyvitamin D [25(OH)D] is associated with greater adiposity and adverse cardiometabolic risk profile. The evidence is inconsistent among South Asian Indians. We aimed to examine associations between 25(OH)D and cardiovascular (CVD) risk markers in a rural and urban cohort from South India.

### Subjects/Methods

In this cross sectional study, 373 individuals (men, n = 205) underwent detailed CVD risk marker assessment including anthropometry [body mass index (BMI), waist, (WC) and hip circumferences (HC)], body composition analysis using dual energy x-ray absorptiometry (DXA), blood pressure and biochemical analysis (glucose, insulin and lipids). The distribution of CVD risk factors were compared across serum 25(OH)D levels, stratified as deficiency (<20 ng/ml), insufficiency (20 to 29 ng/ml) and normal (≥30 ng/ml) levels. Multiple regression analysis, adjusting for potential confounders, was used to study associations of 25(OH)D with adiposity and cardiometabolic traits.

### Results

The mean and standard deviation (SD) of age, BMI and 25(OH)D levels were 41.4 (1.1) years, 25.5 (4.8) kg/m² and 23.4 (10.4) ng/ml respectively. The prevalence of 25(OH)D deficiency was 39.9% in this cohort. Individuals in the 25(OH)D deficiency category had significantly higher mean (SD) BMI [26.6 (5.1) kg/m²], waist circumference [89.9 (12.5) cm] and total fat mass [20.6 (7.9) kg] compared with the Vitamin D sufficient group [BMI: 24.0 (4.4); WC 84.7 (12.0); total fat mass: 15.2 (6.8)]. Significantly inverse associations were observed with DXA measured total and regional fat depots with 25(OH)D levels, while anthropometric

**Funding:** The study was jointly funded by the Indian Institute of Public Health, Hyderabad (WTP Project grant/09-2012) and internal research funds from Christian Medical College, Vellore.

**Competing interests:** The authors have declared that no competing interests exist.

indices of adiposity showed significant inverse association only in women. After adjusting for total fat mass, no significant associations were observed between 25(OH)D and the cardiometabolic traits.

## Conclusions

Our results confirm that lower 25(OH)D is independently associated with both total and regional adiposity, but not with cardiometabolic traits, in this population.

## Introduction

Obesity and 25-hydroxyvitamin D [25(OH)D] deficiency are potentially important modifiable risk factors for cardiometabolic disease, and the prevalence of both conditions is increasing world-wide [1]. Serum concentrations of 25(OH)D have been shown to be inversely related to several cardiovascular disease (CVD) risk traits such as obesity, insulin resistance (IR), type 2 diabetes (T2D), hypertension and cancer [2,3] in different ethnic groups [4–7], while evidence among Asian Indians is inconsistent. A twin burden of higher prevalence of 25(OH)D deficiency [8,9] and increased susceptibility to cardiometabolic risk at lower levels of adiposity is well documented among Asian Indians.

Evidence of a relationship between 25(OH)D deficiency and obesity arises from cross-sectional studies, in which causality cannot be confirmed. A recent systematic review of randomized controlled trials of 25(OH)D supplementation concluded that there is no proven benefit in terms of weight loss, thereby disproving a causal relationship between serum 25(OH)D concentrations and adiposity [10]. Vimaleswaran *et al*, by using a bi-directional mendelian randomization study, concluded that higher BMI leads to lower circulating 25(OH)D levels (the latter falling by 4.2% per 10% increase in BMI) while lower 25(OH)D made little or no contribution to higher BMI [11]. Studies that utilized more robust measurements of adiposity have shown that 25(OH)D levels are inversely associated with total fat percentage and visceral adipose tissue (VAT) in the white Caucasian population [4,12].

Lower 25(OH)D levels could increase cardiovascular disease (CVD) risk by activating the pro-inflammatory cascade, thereby leading to endothelial dysfunction and increased arterial stiffness [13]. However, randomized controlled trials of 25(OH)D supplementation have shown no reduction in CVD risk [14–16]. Studies in South Asian Indians have shown no consistent associations with 25(OH)D levels and cardiometabolic risk markers, and suggest that associations if any, were confounded by obesity [17,18].

Given that regional fat depots demonstrate varying risk associations with cardio-metabolic disease [19], we hypothesized that the relationship between 25(OH)D and adiposity may differ between regional fat depots and total fat mass. Therefore, the current study was undertaken in South Asian Indians with two aims: i) To examine the relationship of 25(OH)D concentration with overall and regional adiposity measured using anthropometry and DXA; ii) To examine associations between 25(OH)D and obesity-related cardiometabolic traits.

## Materials and methods

### Study population

This cross-sectional study included 373 individuals, selected randomly from the Vellore Birth Cohort (VBC) in Vellore (latitude 12.9°N) town, Tamil Nadu, India, recruited from June 2013 –July 2014. Details of the VBC have been described elsewhere [20]. In brief, the VBC was is

population-based birth cohort that was established between 1969–73 and included participants from the urban Vellore town (41%) and adjoining rural villages (59%), thereby representing different socio-economic strata. For the current study (Phase-6 adult follow-up), a total of 1,080 subjects participated, and data on 25(OH)D status were available for 373 individuals [Rural: 182 (48.8%); Urban 191(51.2%)]. All participants underwent a detailed clinical assessment which included the measurement of blood pressure, body composition using anthropometry and DXA, and biochemical CVD risk markers.

## Anthropometric and DXA measurements

Weight was measured with participants in light clothing using a digital weighing scale to the nearest 0.1 kg; height was measured to nearest 1 mm, using a Harpenden portable stadiometer (Holtain Ltd, Crymych, Dyfed, Wales); waist circumference (WC) was measured to the nearest 1 mm, midway between the costal margin and iliac crest in expiration; and hip circumference (HC) was measured to the nearest 1 mm, at the widest part of the buttocks using a non-stretchable tape. An average of three measurements was used for all the anthropometric measures. Blood pressure (BP) was recorded seated using a digital sphygmomanometer (Omron M3 Corporation, Tokyo, Japan) after five minutes seated. Whole-body composition was measured on a Hologic DXA scanner in the fan beam mode (QDR-4500 Discovery A; Hologic, Waltham, MA, USA) using the software provided by the manufacturer (QDR for Windows Version 11.1.2). The regions of interest were automatically defined according to standard procedures as described in the Hologic user's guide. as follows: 1) Android fat: the area of the abdomen included between the line joining the two superior iliac crests and extended cranially up to 20% of the distance between this line and the base of the skull; 2) Gynoid fat: the portion of the legs from the greater trochanter of the femur, extending caudally up to the mid-thigh. Visceral adipose tissue (VAT) was measured in a 5 cm wide region immediately above the iliac crest at a level that approximately coincided with the 4th lumbar vertebrae. Estimated VAT in kg is obtained using an in-built algorithm, which is described in detail elsewhere [21].

## Biochemical measurements

Following a minimum 8 hour overnight fast, venous blood (2ml) was collected in an EDTA fluoride vacutainer tube for fasting glucose and lipids and another sample was drawn 2 hours after a 75g oral anhydrous glucose load. Samples were centrifuged (2,750 rpm for 10 minutes), aliquoted and transported on ice to the central lab for analysis. Samples for parathyroid hormone (PTH) were collected in EDTA tubes, placed on ice and were analysed within 72 hours of venepuncture. Samples for PTH were collected in EDTA tubes on ice. Plasma was separated in refrigerated centrifuge at 4°C and analysed immediately or frozen till the following day at -20°C. PTH assay was performed in Siemens Advia Centaur XP. Plasma glucose and lipids, were measured using enzymatic methods; and serum insulin by a chemiluminescence immunoassay (CLIA). Serum 25(OH)D was measured using a chemiluminescence immunoassay on a Immulite 2000 analyzer. Serum calcium and albumin were measured using a colorimetric method on a Beckman Coulter analyser (Beckman Coulter AU 5800).

Information about physical activity (PA) and socioeconomic status (SES) were obtained using standardized questionnaires administered by trained field workers. Sociodemographic information was collected using standardized questionnaires and included information on educational status, smoking and alcohol consumption, physical activity (PA) and socio-economic status (SES). PA was calculated based on time spent in different types of work, domestic tasks, leisure activities and in walking and cycling with or without a load. Time periods for each activity were multiplied by metabolic constants derived from published tables of the

relative energy expenditure of each task, and summed to create a final PA score. The study was approved by the ethics committee of CMC, Vellore and all participants provided written informed consent.

## Outcome definitions

We used the Endocrine Society clinical practice guidelines to define vitamin D status as: i) Deficiency (<20 ng/mL); ii) Insufficiency (21 to 29 ng/mL); and iii) Sufficiency, (>30 ng/mL) [22]. Cardiometabolic outcomes included impaired fasting glucose, defined as a fasting glucose concentration between 5.6 mmol/l and 7 mmol/l andT2D, defined as either a fasting glucose ≥7 mmol/l and/or a 2-hour glucose ≥11 mmol/l or on treatment for diabetes; based on WHO criteria [23]; hypertension defined as a systolic blood pressure ≥140 mmHg or diastolic blood pressure ≥90 mmHg or on treatment with antihypertensives [24]; and hypertriglyceridemia, defined as a serum triglyceride ≥1.69 mmol/l. HOMA-IR and HOMA-B were calculated using the online calculator (http://www.dtu.ox.ac.uk/homa).

## Statistical methods

Data are presented as mean and standard deviation (SD) for normally distributed variables, median and interquartile range (IQR) for skewed variables and frequency (percentage) for categorical variables. The descriptive characteristics of the study participants were compared across the groups of 25(OH)D status stratified by sex, and p-values were obtained using one-way ANOVA or the non-parametric Kruskall Wallis test, as applicable. Pearson's correlations were used to examine relationships between 25(OH)D and body composition meausres. We used multiple linear regression to study associations of 25(OH)D with body composition variables and cardio-metabolic traits, adjusted for age, area of residence (rural/urban), SES and PA. Estimates are presented as unstandardized β-coefficients and 95% confidence intervals (CI). Multiple logistic regression was used to assess associations between 25(OH)D and binary cardiometabolic outcomes, adjusted for age, area of residence (rural/urban), SES and PA in model 1 and additionally adjusted for total fat mass in model 2. Estimates are presented as odds ratios (OR) and 95% CI. All tests were two-sided at $\alpha = 0.05$ level of significance. A sensitivity analysis was performed, comparing the characteristics of individuals who were included in this analysis with those not included in the current analysis due to the absence of Vitamin D measurements. All analyses were carried out using SPSS Version 21.0 (Armonk, NY: IBM Corp).

## Results

The 373 participants (Males 205; Females 168) were aged between 39 and 44 years. Approximately 50% were urban dwellers. **Table 1** describes the summary statistics of demographic and clinical stratified by sex. Women had significantly higher BMI and total and regional adiposity than men.

Forty percent of the cohort was 25(OH)D deficient and a further 35% were 25(OH)D insufficient (**Table 2**). Men and women in the deficient group had a higher BMI [Mean (SD) Men: 25.4 (4.7), Women 27.7 (5.2)] A decreasing trend was observed in BMI with increasing 25 (OH)D status in both men (BMI p = 0.03) and women (BMI p = 0.002). The prevalence of central obesity (mean WC >80 cm in women and >90 cm in men) also decreased also with increasing 25(OH)D status. DXA-measured total fat mass was significantly higher in the Vitamin D deficient group when compared with insufficient and sufficient groups in both men (p<0.001) and women (p = 0.002). The mean (SD) values of DXA-measured regional adiposity depots were higher in the 25(OH)D deficient group.

**Table 1. Clinical characteristics of study participants.**

|  | Men (n = 205) | Women (n = 168) | P value |
|---|---|---|---|
| Urban[b] | 105 (51.2) | 86 (51.2) | 0.99 |
| Age (years) | 41.4 (1.0) | 41.4 (1.0) | 0.88 |
| BMI (kg/m$^2$) | 24.9 (4.4) | 26.1 (5.2) | 0.02 |
| Physical activity score [a] | 1200 (845–1569) | 1724 (1368–2119) | <0.001 |
| SES (Household possession score)[b] |  |  |  |
| 1 | 54 (26.4) | 51 (30.4) | 0.79 |
| 2 | 46 (22.4) | 36 (21.4) |  |
| 3 | 56 (27.3) | 40 (23.8) |  |
| 4 | 49(23.9) | 41 (24.4) |  |
| Waist Circumference (cm) | 90.9 (12.3) | 83.7 (12.0) | <0.001 |
| Hip Circumference (cm) | 91.7 (8.7) | 95.5 (10.9) | <0.001 |
| Systolic Blood Pressure (mmHg) | 127.0 (16.4) | 118.6 (13.4) | <0.001 |
| Diastolic Blood Pressure (mmHg) | 80.2 (12.9) | 75.8 (9.7) | <0.001 |
| *DXA body composition* |  |  |  |
| Total Fat (kg) | 15.5 (6.4) | 21.4 (7.5) | <0.001 |
| Android Fat (kg) | 1.41 (0.7) | 1.51 (0.7) | 0.19 |
| Gynoid Fat (kg) | 2.5 (0.9) | 3.7 (1.1) | <0.001 |
| Visceral fat (kg) | 0.44 (0.2) | 0.36 (0.2) | <0.001 |
| Leg fat (kg) | 2.9 (1.2) | 4.4 (1.4) | <0.001 |
| Total lean muscle mass (kg) | 53.3 (8.3) | 40.1 (7.4) | <0.001 |
| *Biochemical analysis* |  |  |  |
| HDL—cholesterol (mmol/l) | 1.05 (0.3) | 1.15 (0.2) | <0.001 |
| LDL–cholesterol (mmol/l) | 3.00 (0.8) | 2.86 (0.8) | 0.12 |
| Total Cholesterol(mmol/l) | 4.7 (1.1) | 4.5 (0.9) | 0.04 |
| Triglycerides (mmol/L)[a] | 1.6 (1.0–2.3) | 1.1 (0.7–1.5) | <0.001 |
| Fasting plasma glucose (mmol/L) | 6.1 (2.4) | 5.9 (1.8) | 0.31 |
| Fasting insulin (mlU/ml)[a] | 8.1 (4.5–12.9) | 7.6 (4.5–11.8) | 0.38 |
| HOMA IR[a] | 2.0 (1.1–3.3) | 1.9 (1.0–3.2) | 0.42 |
| HOMA B[a] | 86.1 (42.8–124.4) | 76.0 (46.8–98.9) | 0.13 |
| Vitamin D | 24.6 (10.8) | 21.8 (9.7) | 0.008 |
| *Binary cardiometabolic outcomes* |  |  |  |
| Impaired fasting glucose[b] | 47 (22.9) | 43 (25.6) | 0.55 |
| Type 2 diabetes[b] | 43 (21.0) | 31 (18.5) | 0.54 |
| Hypertension[b] | 45 (22.0) | 12 (7.1) | <0.001 |
| Hypertriglyceridemia[b] | 92 (44.9) | 28 (16.7) | <0.001 |

Values are mean (SD) for normally distributed variables and p value is obtained from t tests.

[a]median (interquartile range) for skewed variables and p value is obtained from non-parametric Mann Whitney test

[b]n (%) for categorical variables and p value is obtained from Pearson Chi-square test.

A significant decreasing trend was observed for total cholesterol (p = 0.04), fasting insulin (p = 0.007) and HOMA-IR (p = 0.01) in men, DBP in women (p = 0.01) and LDL-cholesterol in both sexes (men p = 0.009 and women p = 0.01).

Correlations between 25(OH)D levels and adiposity indices in men and women are shown in **Fig 1**. Consistently negative correlations, ranging from -0.16 to -0.32 were observed between 25(OH)D levels and both DXA-measured fat depots (total and regional) and anthropometric adiposity indices (WC, HC and BMI) in both sexes.

**Table 2. Characteristics of study sample according to Vitamin D status.**

| | 25 (OH) D Status | | | | | | P value[†] | P value[‡] |
|---|---|---|---|---|---|---|---|---|
| | Deficiency (<20 ng/ml) | | Insufficiency (20–29 ng/ml) | | Sufficiency (≥30 ng/ml) | | | |
| | n = 149 | | n = 130 | | n = 94 | | | |
| | Men | Women | Men | Women | Men | Women | | |
| n (%) | 75 (50.3) | 74 (49.7) | 70 (53.8) | 60 (46.2) | 60 (63.8) | 34 (36.2) | | |
| Age (years)[a] | 41.0 (40.0–42.0) | 41.0 (40.8–42.0) | 41.5 (41.0–42.0) | 41.0 (40.0–42.0) | 41.0 (41.0–42.0) | 42.0 (41.0–43.0) | 0.49 | 0.039 |
| Urban[b] | 49 (65.3) | 49 (66.2) | 35 (50.0) | 28 (46.7) | 21 (35.0) | 9 (26.5) | <0.001 | <0.001 |
| Systolic blood pressure (mmHg) | 127 (18) | 120 (14) | 127 (15) | 115 (13.3) | 126 (14) | 118 (11) | 0.91 | 0.07 |
| Diastolic blood pressure (mmHg) | 79 (14) | 78 (9) | 80 (12) | 73 (9) | 80 (12) | 74 (8) | 0.92 | 0.01 |
| **Adiposity markers** | | | | | | | | |
| BMI (kg/m2) | 25.4 (4.7) | 27.7 (5.2) | 25.4 (3.8) | 24.9 (5.0) | 23.6 (4.3) | 24.7 (4.6) | 0.03 | 0.002 |
| Waist circumference (cm) | 92.9 (12.7) | 87.0 (11.6) | 92.3 (11.0) | 81.1 (12.4) | 86.7 (12.3) | 81.2 (10.7) | 0.007 | 0.005 |
| Hip circumference (cm) | 93.7 (9.7) | 98.9 (10.7) | 92.3 (6.9) | 93 (10.6) | 88.5 (8.5) | 92.4 (6.5) | 0.002 | 0.001 |
| Total fat (kg)[a] | 16.6 (11.9–20.3) | 23.5 (18.6–27.2) | 15.5 (11.5–18.9) | 18.9 (13.5–24.1) | 13.4 (8.8–15.9) | 18.6 (12.7–23.5) | <0.001 | 0.002 |
| Android fat (kg)[a] | 1.6 (1.1–2.0) | 1.6 (1.1–2.2) | 1.4 (1.1–1.7) | 1.2 (0.87–1.65) | 1.2 (0.63–1.48) | 1.3 (0.76–1.61) | 0.001 | 0.003 |
| Gynoid fat (kg)[a] | 2.7 (2.0–3.4) | 4.0 (3.1–4.7) | 2.5 (1.9–3.1) | 3.2 (2.5–4.2) | 2.1 (1.56–2.50) | 3.2 (2.65–4.08) | <0.001 | 0.001 |
| Visceral fat (kg)[a] | 0.5 (0.4–0.6) | 0.4 (0.3–0.5) | 0.5 (0.35–0.58) | 0.3 (0.19–0.36) | 0.4 (0.23–0.49) | 0.3 (0.20–0.41) | 0.002 | 0.005 |
| Leg fat (kg) [a] | 3.2 (2.3–4.2) | 4.7 (3.6–5.7) | 2.6 (1.9–3.4) | 3.7 (2.9–4.7) | 2.3 (1.8–2.9) | 3.7 (3.3–5.0) | 0.001 | 0.003 |
| Total lean muscle mass (kg)[a] | 55.0 (47.8–61.4) | 42.4 (36.9–47.1) | 55.2 (49.8–58.8) | 37.8 (34.0–43.5) | 50.9 (43.92–57.52) | 37.5 (34.24–42.44) | 0.12 | 0.004 |
| BMD Hip (g/cm²) | 0.9 (0.1) | 0.9 (0.1) | 0.9 (0.1) | 0.9 (0.1) | 0.9 (0.1) | 0.9 (0.1) | 0.50 | 0.47 |
| BMD Spine (g/cm²) | 1.0 (0.1) | 1.0 (0.1) | 0.9 (0.1) | 1.0 (0.1) | 0.9 (0.1) | 0.9 (0.1) | 0.62 | 0.73 |
| **Biochemical analysis** | | | | | | | | |
| HDL-cholesterol (mmol/L) | 1.01 (0.20) | 1.14 (0.23) | 1.06 (0.31) | 1.14 (0.23) | 1.10 (0.28) | 1.16 (0.25) | 0.11 | 0.92 |
| LDL-cholesterol (mmol/L) | 3.00 (0.83) | 3.05 (0.85) | 3.21 (0.89) | 2.65 (0.71) | 2.75 (0.76) | 2.87 (0.67) | 0.009 | 0.01 |
| Total cholesterol (mmol/L) | 4.61 (1.04) | 4.60 (1.02) | 4.90 (1.09) | 4.28 (0.79) | 4.43 (0.97) | 4.42 (0.78) | 0.04 | 0.13 |
| Triglycerides (mmol/L)[a] | 1.70 (1.17–2.56) | 1.16 (0.83–1.50) | 1.59 (1.02–2.18) | 0.98 (0.71–1.43) | 1.33 (0.88–2.26) | 1.09 (0.69–1.45) | 0.17 | 0.34 |
| Fasting plasma glucose (mmol/L) | 6.02 (1.70) | 6.11 (2.12) | 6.21 (2.73) | 5.78 (1.47) | 6.09 (2.65) | 5.57 (1.67) | 0.89 | 0.32 |
| Fasting insulin (mlU/ml)[a] | 8.8 (5.9–15.4) | 8.2 (5.4–12.2) | 8.9 (6.9–12.1) | 7.5 (4.5–11.7) | 6.14 (2.51–10.7) | 6.07 (4.0–10.6) | 0.007 | 0.13 |
| HOMA-IR[a] | 2.2 (1.6–3.8) | 2.1 (1.2–3.6) | 2.2 (1.4–3.2) | 1.8 (1.0–3.1) | 1.4 (0.6–2.9) | 1.4 (1.0–2. 6) | 0.01 | 0.08 |
| HOMA-B[a] | 90.0 (62.6–128.5) | 84.4 (52.5–104.8) | 89.2 (45.5–127.1) | 65.7 (45.5–114.6) | 62.3 (38.0–114.8) | 65.9 (45.1–80.9) | 0.09 | 0.15 |
| Albumin (g%) | 4.5 (0.3) | 4.3 (0.3) | 4.5 (0.2) | 4.2 (0.3) | 4.5 (0.3) | 4.3 (0.2) | 0.88 | 0.16 |
| Calcium (mg%) | 9.2 (0.5) | 8.9 (0.5) | 9.1 (0.5) | 8.9 (0.4) | 9.3 (0.4) | 9.2 (0.5) | 0.66 | 0.06 |
| PTH (pg/ml)[a] | 41.3 (30.8–55.8) | 43.7 (33.4–61.2) | 42.5 (28.1–54.7) | 39.4 (29.9–55.8) | 35.2 (26.4–46.1) | 42.9 (30.1–56.4) | 0.12 | 0.22 |
| **Cardio-metabolic endpoints** | | | | | | | | |
| Impaired fasting glucose[b] | 18 (24) | 18 (24) | 15 (21) | 18 (30) | 14 (23) | 7 (21) | 0.91 | 0.86 |
| Type 2 diabetes[b] | 18 (24) | 17 (23) | 14 (20) | 9 (15) | 11 (18) | 5 (15) | 0.41 | 0.23 |
| Hypertension[b] | 16 (21) | 7 (10) | 15 (21) | 3 (5) | 14 (23) | 2 (6) | 0.79 | 0.40 |
| Hypertrigleceridemia [b] | 39 (52) | 13 (18) | 32 (46) | 9 (15) | 21 (35) | 6 (18) | 0.05 | 0.93 |

Values are mean (SD) for normally distributed variables and p values obtained from parametric one-way ANOVA

[a]median (interquartile range) for skewed variables and p values obtained from non-parametric Kruskal Wallis test

[b]n (%) for categorical variables and p values obtained from Chi-square trend analysis.

[†] p value for men and

[‡] p value for women. PTH: Parathyroid hormone, HOMA-IR: HOMA-insulin resistance.

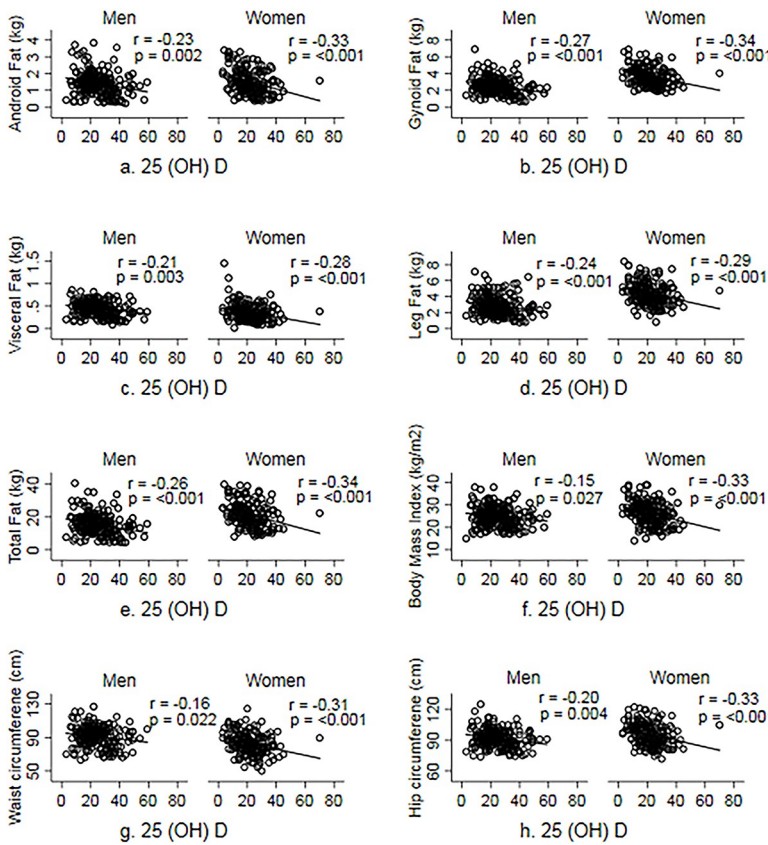

**Fig 1. Correlation between serum 25(OH) D and measures of adiposity.**

After adjustment for age, area of residence, PA and SES, a 1 SD increase in 25(OH)D was associated with a lower total fat mass in men (β = -0.12 kg; 95%CI -0.21, -0.04; p = 0.004) and women (β = -0.20 kg; 95%CI -0.32, -0.08, p = 0.001). Similar inverse and significant association were observed with regional fat depots in both sexes (**Table 3**). Although the associations with

**Table 3. Sex-stratified association between 25(OH)D levels and adiposity traits.**

| Adiposity markers | MEN | | WOMEN | |
|---|---|---|---|---|
| | β (95%CI) | P value | β (95%CI) | P value |
| **Anthropometry** | | | | |
| BMI (kg/m$^2$) | -0.05 (-0.10, 0.01) | 0.10 | -0.13 (-0.22, -0.05) | 0.002 |
| Waist circumference (cm) | -0.12 (-0.31, 0.02) | 0.08 | -0.27 (-0.46, -0.08) | 0.007 |
| Hip circumference (cm) | -0.13 (-0.25, -0.02) | 0.02 | -0.25 (-0.42, -0.08) | 0.005 |
| **DXA** | | | | |
| Total fat (kg) | -0.12 (-0.21, -0.04) | 0.004 | -0.20 (-0.32, -0.08) | 0.001 |
| Android fat (kg) | -0.01 (-0.02, -0.002) | 0.010 | -0.02 (-0.03, -0.008) | 0.001 |
| Visceral fat (kg) | -0.003 (-0.005, -0.001) | 0.010 | -0.005 (-0.008, -0.001) | 0.004 |
| Gynoid fat (kg) | -0.02 (-0.03, -0.008) | 0.002 | -0.03 (-0.05, -0.01) | 0.002 |
| Leg fat (kg) | -0.02 (-0.03, -0.001) | 0.04 | -0.03 (-0.05,-0.004) | 0.02 |

Data are presented as β (unstandardized regression coefficient) with 95% confidence intervals (CI). β represents the mean change in an adiposity marker per unit change in vitamin D. Models were adjusted for age, area of residence, physical activity, socioeconomic status and vitamin D.

**Table 4. Associations between 25(OH) D levels and cardiometabolic traits, stratified by sex.**

| Cardiometabolic traits (outcome variables) | MEN | | | | WOMEN | | | |
|---|---|---|---|---|---|---|---|---|
| | MODEL 1 | | MODEL 2‡ | | MODEL 1 | | MODEL 2‡ | |
| | β (95%CI) | P value | β (95%CI) | P value | β (95%CI) | P value | β (95%CI) | P value |
| Systolic blood pressure (mmHg) | -0.02 (-0.24, 0.19) | 0.85 | 0.06 (-0.15, 0.28) | 0.57 | -0.13 (-0.35, 0.08) | 0.23 | -0.006 (-0.22, 0.21) | 0.96 |
| Diastolic blood pressure (mmHg) | -0.02 (-0.19, 0.14) | 0.79 | 0.06 (-0.10, 0.22) | 0.49 | -0.20 (-0.36, -0.05) | 0.01 | -0.12 (-0.27, 0.03) | 0.12 |
| HDL—cholesterol (mmol/l) | 0.002 (-0.001, 0.006) | 0.19 | 0.002 (-0.002, 0.01) | 0.41 | 0.001 (-0.003, 0.005) | 0.74 | -0.001 (-0.005, 0.003) | 0.69 |
| LDL–cholesterol (mmol/l) | -0.01 (-0.02, 0.00) | 0.05 | -0.01 (-0.02, 0.002) | 0.09 | -0.008 (-0.02, 0.005) | 0.23 | -0.002 (-0.02, 0.01) | 0.78 |
| Total cholesterol (mmol/l) | -0.01 (-0.03, 0.002) | 0.10 | -0.01 (-0.02,0.004) | 0.17 | -0.009 (-0.02, 0.005) | 0.21 | -0.005 (-0.02, 0.01) | 0.54 |
| Triglyceride (mmol/l) | -0.01 (-0.03, 0.01) | 0.51 | -0.004 (-0.02, 0.01) | 0.73 | -0.005 (-0.04, 0.03) | 0.75 | 0.001 (-0.03, 0.03) | 0.96 |
| Insulin (uIU/ml) | -0.12 (-0.24, 0.01) | 0.06 | -0.01 (-0.13, 0.09) | 0.74 | -0.05 (-0.15, 0.04) | 0.29 | 0.03 (-0.04, 0.12) | 0.36 |
| Fasting glucose (mmol/l) | -0.004 (-0.04, 0.03) | 0.78 | 0.001 (-0.03, 0.03) | 0.94 | -0.02 (-0.04, 0.01) | 0.18 | -0.02 (-0.05, 0.01) | 0.27 |
| HOMA-IR | -0.03 (-0.08, 0.01) | 0.19 | -0.003 (-0.05, 0.04) | 0.91 | -0.02 (-0.05, 0.01) | 0.18 | 0.003 (-0.03, 0.03) | 0.81 |
| HOMA-B | -0.11 (-0.27, 0.06) | 0.22 | -0.12 (-0.29, 0.05) | 0.16 | -0.06 (-0.19, 0.08) | 0.43 | -0.08 (-0.23, 0.06) | 0.25 |
| *Binary Cardiometabolic outcomes[a]* | OR (95% CI) | | OR (95% CI) | | OR (95% CI) | | OR (95% CI) | |
| Impaired fasting glucose | 0.9 (0.5, 2.0) | 0.94 | 0.8 (0.4, 1.7) | 0.55 | 1.4 (0.5, 3.7) | 0.49 | 1.2 (0.4, 3.6) | 0.69 |
| Type 2 diabetes | 1.4 (0.6, 3.1) | 0.40 | 1.3 (0.6, 2.9) | 0.56 | 1.8 (0.6, 5.7) | 0.30 | 2.1 (0.6, 7.4) | 0.23 |
| Hypertension | 0.9 (0.4, 1.9) | 0.78 | 0.8 (0.4, 1.7) | 0.49 | 1.5 (0.3, 7.7) | 0.67 | 1.1 (0.2, 6.3) | 0.88 |
| Hypertriglyceridemia | 1.7 (0.9, 3.2) | 0.12 | 1.5 (0.8, 2.9) | 0.22 | 1.1 (0.4, 3.1) | 0.89 | 0.8 (0.2, 2.4) | 0.66 |

Data are presented as β (unstandardized regression coefficient) with 95% confidence interval (CI). β represents the mean change in a cardiometabolic trait per unit change in vitamin D. Model 1—adjusted for age, area of residence, physical activity and socioeconomic status. Model 2‡—adjusted for age, area of residence, physical activity, socioeconomic status and total fat mass.

[a]Data for binary outcomes are presented as OR (95% CI).

anthropometric indices of adiposity were directionally similar, these were stronger among women than men.

When CVD outcomes were analysed as continuous measurements or and dichotomized end-points, no significant associations were observed with serum 25(OH)D levels and cardio-metabolic risk traits, except for diastolic blood pressure in women (p = 0.01) (**Table 4**).

In our sensitivity analysis, comparison of individuals included the current analysis with those who were not studied due to absent 25(OH)D measurements did not show any signifi-cant differences in most of the characteristics, except that a higher proportion of women in the 'not studied group' had T2D, and men in the studied group had higher glucose and HOMA-IR (**S1 Table**).

## Discussion

Our data show that 25(OH)D deficiency is common in a general population in South India. Levels of 25(OH)D inversely associate with both overall and regional adiposity. The associa-tions of different regional fat depots to 25(OH)D levels were of approximately equal magni-tude, and comparable with that of total fat mass, signifying, contrary to our hypothesis that 25 (OH)D is associated with overall fatness rather than with specific regional fat depots. The absence of significant associations with cardiometabolic risk markers in our study, suggests

that 25(OH)D levels do not have an independent effect on cardiometabolic risk in this population.

The prevalence of 25(OH)D in our population is comparable to other studies from India among people of similar age [8,9,25]. Baker et al, showed a similar inverse relationship between DXA-measured total fat mass and 25(OH)D levels in younger (18–24 years) rural Asian Indians [18] Our study adds to the existing evidence, by showing this relationship in middle aged (41–43 years) rural and urban Indians. Studies in other ethnic groups have shown similar inverse relationships with total fat and VAT, but did not assess other depots [4,26,27].

VAT accumulation is thought to be an important determinant of 25(OH)D status because of its high metabolic activity [27–29]. Studies have shown that interventions leading to VAT loss increases circulating 25(OH)D levels [30]. Our results confirm an inverse relationship between VAT and 25(OH)D levels. However, the similar significant inverse association observed with other fat depots and the narrow range of β estimates suggest that circulating 25(OH)D maybe affected by an overall increase in fat tissue and not related to any specific fat patterning. The consistent inverse linear correlations between regional fat depots and 25(OH)D further support our findings. It is possible that 25(OH)D is buffered by storage in enlarged fat mass irrespective of a specific region, and that global accumulation of excessive fat, rather than a specific fat depot may be an important determinant of 25(OH)D concentration.

Several mechanisms have been proposed for the link between increased adiposity and lower 25(OH)D concentration [31]. These include 1) reduced expression of the CYP24A1 gene, coding for 1-alpha hydroxylase, which is responsible for conversion of vitamin-D to its active form; 2) decreased vitamin D production secondary to reduced vitamin-D receptors in adipocytes [32,33]; 3) increased sequestration in adipocytes and reduced bioavailability [34]; 4) increased 24-hydroxylase activity leading to enhanced catabolism of vitamin D [35]; and secondary hyperparathyroidism leading to increased lipogenesis and volumetric dilution [36].

We acknowledge that the association between obesity and vitamin D is highly debated and It is not clear if a causal relationship exists between greater adiposity and low 25(OH)D levels or, vice versa [36,37]. Nevertheless, our study adds to the existing evidence that greater adiposity is related to lower vitamin D levels in Asian Indians, as in other ethnic groups.

Low 25(OH)D levels have been shown to be associated with insulin resistance, hypertension, CVD and metabolic syndrome in some studies [3,38,39]. However, experimental and human intervention trials of vitamin D supplementation have shown inconsistent or inconclusive effects on CVD risk markers[40,41]. Several mechanisms have been proposed to explain increased CVD risk with low 25(OH)D levels [42]. Our results align with studies that epidemiological showed no association between 25(OH)D levels and CVD risk [17]. Furthermore, the attenuation of significant relationships between 25(OH)D levels and some CVD traits (hypertriglyceridemia, diastolic blood pressure) after adjusting for total fat mass suggests that any relationship between 25(OH)D and CVD risk markers maybe driven entirely by adiposity.

This is the first study to examine the relationship of several adiposity indices measured using anthropometry and, absolute fat mass quantified using DXA, as well as a range of cardiometabolic risk factors, with 25(OH)D concentration, in Asian Indians. We used total fat mass to account for confounding effects of adiposity, while most previous studies used BMI. A limitation was that our cohort participants were predominantly in the overweight or obese category, and therefore generalization to normal weight individuals may not be possible and warrants future investigation. Another limitation was that we had not have data on sunlight exposure, or dietary vitamin D or calcium intake, which have been previously shown attenuate associations in other studies [43]. However, we have accounted for other important confounders (age, physical activity, socio-economic status and total fat mass). We used chemiluminescence immunoassay for PTH analysis, rather than the gold standard Liquid chromatography-

tandem mass spectroscopy since this was not possible in our clinical setting. Given the robustness of the platform used, we do not anticipate any differences in the results. We do not believe that our study was affected by bias, and the participants were similar in most baseline characteristics compared to the rest of the cohort.

In conclusion, our results confirm an inverse association between 25(OH)D levels and both total and regional adiposity in South Asian Indians, and adds to existing evidence that 25(OH) D is an important predictor of overall adiposity and of a specific body fat pattern. The absence of association between 25(OH)D and a range of CVD risk factors suggests that the associations of the 25(OH)D with CVD risk may be largely be confounded by adiposity.

## Supporting information

**S1 Table. Sensitivity analysis comparing individuals studied vs. not studied (without 25 (OH)D measurements).**
(DOCX)

## Acknowledgments

The authors would like to thank the field team of the Vellore birth cohort for data collection.

## Author Contributions

**Conceptualization:** Belavendra Antonisami, Senthil K. Vasan, Nihal Thomas, Thomas V. Paul.

**Data curation:** Reka Karuppusami, Belavendra Antonisami, Mahasampath Gowri, Hepsy Y. Selliah.

**Formal analysis:** Reka Karuppusami, Belavendra Antonisami, Mahasampath Gowri, Hepsy Y. Selliah.

**Funding acquisition:** Nihal Thomas, Thomas V. Paul.

**Investigation:** Senthil K. Vasan, Geethanjali Arulappan.

**Methodology:** Reka Karuppusami, Belavendra Antonisami, Senthil K. Vasan, Mahasampath Gowri, Nihal Thomas, Thomas V. Paul.

**Project administration:** Felix Jebasingh, Nihal Thomas, Thomas V. Paul.

**Resources:** Belavendra Antonisami, Geethanjali Arulappan, Felix Jebasingh, Nihal Thomas, Thomas V. Paul.

**Software:** Belavendra Antonisami.

**Supervision:** Senthil K. Vasan, Nihal Thomas, Thomas V. Paul.

**Validation:** Senthil K. Vasan, Thomas V. Paul.

**Writing – original draft:** Reka Karuppusami, Senthil K. Vasan.

**Writing – review & editing:** Reka Karuppusami, Belavendra Antonisami, Senthil K. Vasan, Mahasampath Gowri, Hepsy Y. Selliah, Geethanjali Arulappan, Felix Jebasingh, Nihal Thomas, Thomas V. Paul.

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
