## [Decision Letter · Decision Letter 0]

16 Oct 2020

PONE-D-20-29111

Serum 25-Hydroxy vitamin D levels are inversely associated with total and regional adiposity and not with cardiometabolic traits in South Asian Indians

PLOS ONE

Dear Dr. PAUL,

Thank you for submitting your manuscript to PLOS ONE. After careful consideration, we feel that it has merit but does not fully meet PLOS ONE’s publication criteria as it currently stands. Therefore, we invite you to submit a revised version of the manuscript that addresses the points raised during the review process.

We look forward to receiving your revised manuscript.

Kind regards,

Mauro Lombardo

Academic Editor

PLOS ONE

Journal Requirements:

Reviewers' comments:

Reviewer's Responses to Questions

**Comments to the Author**

1. Is the manuscript technically sound, and do the data support the conclusions?

Reviewer #1: Yes

Reviewer #2: Yes

Reviewer #3: Partly

2. Has the statistical analysis been performed appropriately and rigorously? 

Reviewer #1: Yes

Reviewer #2: Yes

Reviewer #3: Yes

3. Have the authors made all data underlying the findings in their manuscript fully available?

Reviewer #1: Yes

Reviewer #2: Yes

Reviewer #3: No

4. Is the manuscript presented in an intelligible fashion and written in standard English?

Reviewer #1: Yes

Reviewer #2: Yes

Reviewer #3: Yes

5. Review Comments to the Author

Reviewer #1: This research paper is rigorously done and well written.

Comments:

#1 Regarding the title, as American Journal of Cardiology editor William Roberts said, “If the manuscript’s message is in the title, the reader’s curiosity may be immediately satisfied and the next article then sought.” You might consider editing the title.

#2 You wrote "hypertension defined as a systolic blood pressure ≥140 mmHg or diastolic blood pressure ≥90 mmHg".

Were the BPs the average of 2 readings in at least 2 consecutive visits to MD at least 1 week apart as per the JACHO guidelines? Or some other guidelines?

Reviewer #2: The article “Serum 25-Hydroxy vitamin D levels are inversely associated with total and regional adiposity and not with cardiometabolic traits in South Asian Indians” has focused on relevant health problems – vitamin D deficiency and it evaluation with anthropometric and cardiometabolic traits, nevertheless there are some issues requiring an improvement:

1. Some descriptions in methodology are not clear. Equipment used for body weight and height measurements could be better described.

2. How much blood was taken an how it was protected for the study?

3. The gold standard for vitamin D determination is the Liquid Chromatography-tandem Mass Spectrometry – it could be included in limitation of the study.

4. English and scientific language should be improved.

5. References should be prepared according to Instructions for Authors of the Journal.

6. In discussion more information about mechanisms connected with vitamin D deficiency and obesity or other metabolic risk factors should be given. It is worth to citate an article written by Pelczyńska M. et al. Hypovitaminosis D and adipose tissue – cause and effect relationships in obesity. Ann Agric Environ Med. 2016.

Reviewer #3: SUMMARY OF THE RESEARCH AND OVERALL IMPRESSION

In this work, the authors intend to address the relationship between serum levels of Vitamin D, and fat distribution in the body, along with the presence of CVD (cardiovascular disease) risk traits, in South Asian Indians. In the introduction section, the authors point to a lack of consistency in data regarding the correlation between vitamin D levels and obesity in Asian Indians (several publications are indicated).

For this reason, the authors designed a cross-sectional study, in order to address the correlation of specific regional fat depots (as opposite to total body fat content) with the levels of Vitamin D which and several CVD risk factors.

The methodology employed by the authors include DXA scans along with anthropometrical, clinical and biochemical parameters in a set of 373 individuals from a particular region in South India.

In their paper, the authors add to the existing evidence that Vitamin D is a predictor of the overall obese phenotype (they found that both men and women with the lowest levels of vitamin D are more obese), ruling out a body fat patterning-related phenotype (no relationship of vitamin D levels with regional fat depots, but instead, with total fat mass). Moreover, the authors found no correlation between vitamin D levels and CVD traits (hypertriglyceridemia, diastolic blood pressure).

The main strengths of this paper is that it addresses the possible role of regional fat depots, and adds more evidence to the association between vitamin D levels and adiposity. It also goes in line with previous data regarding the relationship between Vitamin D levels and CVD risk factors. The manuscript is well organized so it could be accessible to non-specialists. The authors also acknowledge some of the flaws of the work, especially those related to the lack of a dietary record (type of fat consumed for instance) and calcium intake, which could impact heavily in the levels of vitamin D.

SPECIFIC AREAS OF IMPROVEMENT

-MINOR ISSUES: Grammar and spelling issues need to be addressed.

-MAJOR ISSUES: Some of the weaknesses are that some data and analyses are not sufficient to support the claims: This study requires additional information in order to make such conclusions or alternatively, the authors should include more information that clarifies and justifies their choice of the data presented.

1-The size of the sample studied (373 individuals), and the fact of that population being overweight, makes difficult to assume the statement in the title “in South Asian Indians”; maybe the authors could rephrase the title in order to make it more accurate to the data presented.

2-The detailed “clinical assessment” regarding CVD traits/risk factors, includes a single determination of blood pressure, glucose levels and HOMA index. There is no information about medication (proper anamnesis) or the levels of some inflammatory factors (such as: leptin, adiponectin, resistin, TNF, ICAM, CRP, fibrinogen) and arterial stiftness data. Could the authors have access to some of these parameters to reinforce the statement about the relationship between vitamin D levels in this population and cardiovascular disease traits?

3-The authors show a higher value of HOMA-IR in the vitamin D deficient group, however, there is no difference in type 2 diabetes data between the different groups. Maybe the authors could add HOMA-B calculation in this table in order to get an idea of beta cell performance in relation with vitamin D.

4-In the correlation between vitamin D and CVD traits, the authors propose in the discussion section that a plausible cause could be the levels of PTH, but there is no change in such levels in their work. Could the authors extend in this comment?

5-Despite the fact that the authors have provided their analyses stratified by socioeconomic status and physical activity, the authors should be encouraged to disclose the raw data of these variables, in order to have a complete idea of the characteristics of the population studied.

6. PLOS authors have the option to publish the peer review history of their article (what does this mean?). If published, this will include your full peer review and any attached files.

Reviewer #1: **Yes: **David K Cundiff

Reviewer #2: No

Reviewer #3: No

---

## [Author Response · Author response to Decision Letter 0]

17 Nov 2020

Reviewer #1: This research paper is rigorously done and well written.

Comments:

#1 Regarding the title, as American Journal of Cardiology editor William Roberts said, “If the manuscript’s message is in the title, the reader’s curiosity may be immediately satisfied and the next article then sought.” You might consider editing the title.

Response: We have now changed the title as suggested by the editor and the key message is not in the title. The revised title reads as “Association between serum 25-Hydroxy vitamin D and total and regional adiposity and cardiometabolic traits”

#2 You wrote "hypertension defined as a systolic blood pressure ≥140 mmHg or diastolic blood pressure ≥90 mmHg".

Were the BPs the average of 2 readings in at least 2 consecutive visits to MD at least 1 week apart as per the JACHO guidelines? Or some other guidelines?

Response: We apologise for the lack of clarity in the definition of hypertension. Our used the JNC 8 (Eighth Joint National Committee on the Prevention, Detection, Evaluation, and Treatment of High Blood Pressure) definition of hypertension as systolic blood pressure ≥140 mmHg or diastolic blood pressure ≥90 mmHg or on treatment (Ref: James PA., et al 2014 evidence-based guideline for the management of high blood pressure in adults: report from the panel members appointed to the Eighth Joint National Committee (JNC 8). JAMA 2014 311:507–20)). This reference is now cited and definition edited in the revised manuscript. 

Reviewer #2: The article “Serum 25-Hydroxy vitamin D levels are inversely associated with total and regional adiposity and not with cardiometabolic traits in South Asian Indians” has focused on relevant health problems – vitamin D deficiency and it evaluation with anthropometric and cardiometabolic traits, nevertheless there are some issues requiring an improvement:

1. Some descriptions in methodology are not clear. Equipment used for body weight and height measurements could be better described.

Response: We regret the lack of clarity provided with regard to anthropometric measurements and blood pressure. We have now included detailed descriptions of the same under the methods section in Page 6 of the revised manuscript. 

Detailed description added are as below: Weight was measured with participants in light clothing using a digital weighing scale to the nearest 0.1 kg; height was measured to nearest 1 mm, using a Harpenden portable stadiometer (Holtain Ltd, Crymych, Dyfed, Wales); waist circumference (WC) was measured to the nearest 1 mm, midway between the costal margin and iliac crest in expiration; and hip circumference (HC) was measured to the nearest 1 mm, at the widest part of the buttocks using a non-stretchable tape. An average of three measurements were taken for all the anthropometric measures. Blood pressure (BP) was recorded seated using a digital sphygmomanometer (Omron M3 Corporation, Tokyo, Japan) after five minutes seated.

2. How much blood was taken and how it was protected for the study?

Response: Following a minimum of 8 hour overnight fast, venous blood (2ml) was collected in an EDTA fluoride vacutainer tube for fasting glucose and lipids and another sample was drawn after 2 hours following 75g oral glucose tolerance test for 2-hour glucose. Samples were centrifuged (2,750 rpm for 10 minutes), aliquoted and transported on ice to the central lab for analysis. Samples for PTH were collected in EDTA tubes on ice and analysed within 72 hours of venepuncture. Plasma was separated in refrigerated centrifuge at 4OC and analysed immediately or frozen till the following day at -20OC. PTH assay was performed in Siemens Advia Centaur XP.

We have now included a detailed section under ‘Biochemical measurements” which contains all the above information and various platforms used for biochemical assays. 

3. The gold standard for vitamin D determination is the Liquid Chromatography-tandem Mass Spectrometry – it could be included in limitation of the study.

Response: We agree with the reviewer that tandem mass spectrometry is the gold standard. Given the high number of samples and in an epidemiological setting this was not possible in our study. As suggested by the reviewer, we have now included as a limitation of our study. 

4. English and scientific language should be improved.

Response: We have now carefully looked through the manuscript and revised the language as appropriate. 

5. References should be prepared according to Instructions for Authors of the Journal.

Response: References are now changed as per PLoS One journal recommendations. 

6. In discussion more information about mechanisms connected with vitamin D deficiency and obesity or other metabolic risk factors should be given. It is worth to citate an article written by Pelczyńska M. et al. Hypovitaminosis D and adipose tissue – cause and effect relationships in obesity. Ann Agric Environ Med. 2016.

Response: We thank the reviewer for this suggestion. We have now included more information regarding the mechanisms and also cited the suggested article. 

Reviewer #3: SUMMARY OF THE RESEARCH AND OVERALL IMPRESSION

In this work, the authors intend to address the relationship between serum levels of Vitamin D, and fat distribution in the body, along with the presence of CVD (cardiovascular disease) risk traits, in South Asian Indians. In the introduction section, the authors point to a lack of consistency in data regarding the correlation between vitamin D levels and obesity in Asian Indians (several publications are indicated).

For this reason, the authors designed a cross-sectional study, in order to address the correlation of specific regional fat depots (as opposite to total body fat content) with the levels of Vitamin D which and several CVD risk factors.

The methodology employed by the authors include DXA scans along with anthropometrical, clinical and biochemical parameters in a set of 373 individuals from a particular region in South India.

In their paper, the authors add to the existing evidence that Vitamin D is a predictor of the overall obese phenotype (they found that both men and women with the lowest levels of vitamin D are more obese), ruling out a body fat patterning-related phenotype (no relationship of vitamin D levels with regional fat depots, but instead, with total fat mass). Moreover, the authors found no correlation between vitamin D levels and CVD traits (hypertriglyceridemia, diastolic blood pressure).

The main strengths of this paper is that it addresses the possible role of regional fat depots, and adds more evidence to the association between vitamin D levels and adiposity. It also goes in line with previous data regarding the relationship between Vitamin D levels and CVD risk factors. The manuscript is well organized so it could be accessible to non-specialists. The authors also acknowledge some of the flaws of the work, especially those related to the lack of a dietary record (type of fat consumed for instance) and calcium intake, which could impact heavily in the levels of vitamin D.

SPECIFIC AREAS OF IMPROVEMENT

-MINOR ISSUES: Grammar and spelling issues need to be addressed.

Response: We have now carefully looked through the manuscript and revised the language as appropriate

-MAJOR ISSUES: Some of the weaknesses are that some data and analyses are not sufficient to support the claims: This study requires additional information in order to make such conclusions or alternatively, the authors should include more information that clarifies and justifies their choice of the data presented.

1-The size of the sample studied (373 individuals), and the fact of that population being overweight, makes difficult to assume the statement in the title “in South Asian Indians”; maybe the authors could rephrase the title in order to make it more accurate to the data presented.

Response: We agree with the reviewer and have also highlighted this as a limitation in our discussion that the BMI of our participants meet the criteria for “overweight”, and even obese when BMI cut-offs for Asian standards are considered. We have therefore removed “in South Asian Indians” from the title and the revised title reads as “Association between serum 25-Hydroxy vitamin D and total and regional adiposity and cardiometabolic traits”

2-The detailed “clinical assessment” regarding CVD traits/risk factors, includes a single determination of blood pressure, glucose levels and HOMA index. There is no information about medication (proper anamnesis) or the levels of some inflammatory factors (such as: leptin, adiponectin, resistin, TNF, ICAM, CRP, fibrinogen) and arterial stiftness data. Could the authors have access to some of these parameters to reinforce the statement about the relationship between vitamin D levels in this population and cardiovascular disease traits?

Response: We agree with the reviewer that other information as mentioned would add more validity to our results. Unfortunately, we do not have any information on medications (other than anti-hypertensives and anti-diabetic drugs). Also, other inflammatory markers as mentioned by the reviewer were nor measured in our cohort participants. This is added limitation to our study

3-The authors show a higher value of HOMA-IR in the vitamin D deficient group, however, there is no difference in type 2 diabetes data between the different groups. Maybe the authors could add HOMA-B calculation in this table in order to get an idea of beta cell performance in relation with vitamin D.

Response: We thank the reviewer for this suggestion. We have now included information on HOMA-B in Tables 1 and 2. There was no significant difference in HOMA-B amongst the groups studied and we have included this in the results and discussion section. 

4-In the correlation between vitamin D and CVD traits, the authors propose in the discussion section that a plausible cause could be the levels of PTH, but there is no change in such levels in their work. Could the authors extend in this comment?

Response: We thank the reviewer for bringing this up. We have now carefully read through this and we feel it is not appropriate to refer to PTH levels here. Therefore, we have now removed the above statement as there are several mechanisms that underpin the vitamin D-CVD risk association We have now revised the manuscript accordingly and cited relevant references. 

5-Despite the fact that the authors have provided their analyses stratified by socioeconomic status and physical activity, the authors should be encouraged to disclose the raw data of these variables, in order to have a complete idea of the characteristics of the population studied.

Response: We had previously included information on SES and PA in only the supplement table. We have now included this information even in the main tables and also provided detailed descriptions as how information on SES and PA were collected.

---

## [Decision Letter · Decision Letter 1]

30 Nov 2020

Association of serum 25-Hydroxy vitamin D with total and regional adiposity and cardiometabolic traits

PONE-D-20-29111R1

Dear Dr. PAUL,

We’re pleased to inform you that your manuscript has been judged scientifically suitable for publication and will be formally accepted for publication once it meets all outstanding technical requirements.

Kind regards,

Mauro Lombardo

Academic Editor

PLOS ONE

Additional Editor Comments (optional):

Reviewers' comments:

Reviewer's Responses to Questions

**Comments to the Author**

1. If the authors have adequately addressed your comments raised in a previous round of review and you feel that this manuscript is now acceptable for publication, you may indicate that here to bypass the “Comments to the Author” section, enter your conflict of interest statement in the “Confidential to Editor” section, and submit your "Accept" recommendation.

Reviewer #3: All comments have been addressed

2. Is the manuscript technically sound, and do the data support the conclusions?

Reviewer #3: Yes

3. Has the statistical analysis been performed appropriately and rigorously? 

Reviewer #3: Yes

4. Have the authors made all data underlying the findings in their manuscript fully available?

Reviewer #3: Yes

5. Is the manuscript presented in an intelligible fashion and written in standard English?

Reviewer #3: Yes

6. Review Comments to the Author

Reviewer #3: The authors have included a proper response to all the concerns raised in the first round of revision, and for this reason I encourage the publication of this paper.

7. PLOS authors have the option to publish the peer review history of their article (what does this mean?). If published, this will include your full peer review and any attached files.

Reviewer #3: No

---

## [Editor Report · Acceptance letter]

7 Dec 2020

PONE-D-20-29111R1 

Association of serum 25-Hydroxy vitamin D with total and regional adiposity and cardiometabolic traits 

Dear Dr. PAUL:

I'm pleased to inform you that your manuscript has been deemed suitable for publication in PLOS ONE. Congratulations! Your manuscript is now with our production department. 

Kind regards, 

on behalf of

Dr. Mauro Lombardo 

Academic Editor

PLOS ONE